# Enhanced fibrinolysis detection in a natural occurring canine model with intracavitary effusions: Comparison and degree of agreement between thromboelastometry and FDPs, D-dimer and fibrinogen concentrations

**Andrea Zoia**[1]*, **Michele Drigo**[2], **Christine J. Piek**[3], **Helena Calcini**[1], **Marco Caldin**[4], **Paolo Simioni**[5]

1 Division of Internal Medicine, San Marco Veterinary Clinic, Padua, Italy, 2 Department of Medicina Animale, Produzione e Salute, Padua University, Legnaro, Italy, 3 Department of Clinical Sciences of Companion Animals, Faculty of Veterinary Medicine, Utrecht University, Utrecht, The Netherlands, 4 Division of Clinical Pathology, Laboratorio d'Analisi Veterinarie San Marco, Padua, Italy, 5 Department of Cardiologic, Thoracic and Vascular Sciences, University of Padua Medical School, Padua, Italy

* zoia.andrea06@googlemail.com

**Data Availability Statement:** The raw data were extracted from the patients' medical records and

## Abstract

Dogs with intracavitary effusion have coagulative abnormalities indicative of primary fibrinolysis/hyperfibrinolysis. The aim of this case control study was to investigate by rotational thromboelastometry (ROTEM) and standard coagulation tests (fibrin-fibrinogen degradation products, D-dimer and fibrinogen) fibrinolysis in dogs with intracavitary effusions. Thirty-two dogs with intracavitary effusion and 32 control sick dogs without effusion were studied. Frequency of fibrinolysis grade of severity (i.e., hypofibrinolysis/basal fibrinolysis *vs* increased fibrinolysis *vs* hyperfibrinolysis) by ROTEM and standard coagulation tests were compared between groups. Pattern of fibrinolysis by ROTEM (i.e., late *vs* intermediate *vs* fulminant) and type of fibrinolysis by standard coagulation tests (i.e., hypofibrinolysis/basal fibrinolysis *vs* primary fibrinolysis *vs* secondary fibrinolysis *vs* primary hyperfibrinolysis *vs* secondary hyperfibrinolysis) were also compared between groups. Dogs with intracavitary effusion had a lesser degree of hypofibrinolysis and basal fibrinolysis and a higher degree of increased fibrinolysis and hyperfibrinolysis compared to controls, both by ROTEM and by standard coagulation tests ($P = 0.042$ and $P = 0.017$, respectively). Nevertheless, there was a poor agreement between the two classification schemes (34.4%, K = 0.06, 95% CI = -0.14 – +0.26). Dogs with intracavitary effusion showed, by ROTEM, a lesser degree of hypofibrinolysis and basal fibrinolysis and a higher degree of late, intermediate, and fulminant fibrinolysis compared to controls ($P = 0.044$). Finally, dogs with intracavitary effusion had, by standard coagulation tests, a higher frequency of primary fibrinolysis and primary hyperfibrinolysis and a lower frequency of secondary fibrinolysis compared to controls. Dogs with intracavitary effusion showed an increased frequency and a different and more severe pattern of fibrinolysis compared to controls.

therefore cannot be made publicly available. Medical records are protected by the Italian code of personal data protection D.Lgs 196/2003. Nevertheless, no restrictions have been imposed upon sharing an anonymized data set necessary to replicate our study findings. An anonymized data set is available as a Supporting Information file.

**Funding:** The authors received no specific funding for this work. Nevertheless, the corresponding author declares, on behalf of all authors, the following commercial affiliation: "San Marco Veterinary Clinic, Padua, Italy". This funder provided support in the form of salaries for authors [AZ, HC, and MC], but did not have any additional role in the study design, data collection and analysis, decision to publish, or preparation of the manuscript. The specific roles of these authors are articulated in the 'author contributions' section.

**Competing interests:** The corresponding author declares, on behalf of all authors, the following commercial affiliation: "San Marco Veterinary Clinic, Padua, Italy", for authors [AZ, HC, and MC]. This commercial affiliation does not alter our adherence to PLOS ONE policies on sharing data and materials.

## Introduction

Coagulation and fibrinolysis are precisely regulated by the measured participation of substrates, activators, inhibitors, cofactors and receptors. Fibrinolysis is the process whereby stable fibrin strands are broken down by plasmin [1]. Basal fibrinolysis, the ongoing removal of fibrin deposits, ensures blood fluidity while preventing blood loss. Physiologic fibrinolysis is localized fibrinolysis in response to thrombosis, and is necessary for the re-establishment of blood flow [2]. Physiologic fibrinolysis is mediated by fibrin-bound plasmin [2]. Primary fibrinolysis develops independently of intravascular activation of coagulation, and plasmin is formed without concomitant formation of thrombin [3]. It is mediated by plasma-free plasmin. Generalized fibrinogenolysis occurs when the production of plasmin within the general circulation overwhelms the neutralizing capacity of antiplasmins, potentially leading to severe bleeding, a disorder known as primary hyperfibrinolysis [2], primary hyperfibrinogenolysis [2,4,5], or pathologic fibrinolysis [2]. Finally, when fibrinolysis occurs as an appropriate response to persistent thrombin generation it is termed secondary or reactive fibrinolysis. This phenomenon keeps blood vessels patent by resolving redundant clots [6], and it is present in virtually every patient with disseminated intravascular coagulation [2]. Secondary fibrinolysis has also been reported in inflammatory diseases, such as sepsis [2,7,8].

In the clinical setting, having elevated plasma fibrin-fibrinogen degradation products (FDPs) with a normal D-dimer concentration has been suggested as a possible indicator of primary fibrinolysis/hyperfibrinolysis [4,9,10]. We recently demonstrated that dogs with ascites or pleural effusion had abnormalities of their fibrinolytic system indicative of primary fibrinolysis/hyperfibrinolysis, based on coagulation tests [11– 13]. In fact, intracavitary effusions, which have a demonstrated inherently fibrinolytic activity in humans, dogs, and horses [11–21] are continuously exchanged with the systemic circulation [14,22]. Therefore, upon re-entering into the circulatory system the intracavitary fluids might contribute to the enhanced fibrinolysis detected in these dogs, as previously documented to be the case in humans [14,15,23–27].

Fibrinolytic activity can also be assessed by viscoelastic hemostasis analyzers such as rotational thromboelastometry (ROTEM). Rotational thromboelastometry is a point-of-care device that rapidly detects systemic changes in *in vivo* coagulation. In the ROTEM, coagulation is activated with ellagic acid (INTEM test) or tissue factor (EXTEM test). This is carried out to standardize the *in vitro* coagulation process and subsequent fibrinolysis. The latter can be quantified by clot lysis parameters such as the EXTEM lysis index at 60 minutes (LI60) and the EXTEM maximum lysis (ML) [28]. LI60 is the percentage residual clot firmness at 60 minutes after the thromboelastometric coagulation time, which is the time in seconds from the test start until a clot firmness of 2 mm is obtained. ML is the percentage reduction in maximum amplitude of clot firmness reached during the run time [28]. In addition, EXTEM clot amplitude at 5 minutes (A5), clot amplitude firmness 5 minutes after a clot firmness amplitude of 2 mm has been reached, can be used for early detection of fibrinolysis [28], and identifying patients developing hyperfibrinolysis [29].

The aims of this study were to assess the following hypotheses. First, we hypothesized that ROTEM could detect an enhanced and more severe pattern of fibrinolysis in dogs with intracavitary effusions compared to dogs without effusion. Second, we hypothesized that there was an agreement in fibrinolysis severity detection between the combination of ROTEM assay results and fibrinogen concentrations, or alternatively by concentrations of FDPs, D-dimers and fibrinogen. Finally, we also hypothesized that dogs with intracavitary effusions had more primary fibrinolysis/hyperfibrinolysis compared to dogs without effusion when assessed by concentrations of FDPs, D-dimers and fibrinogen.

## Material and methods

### Animals

In this case control study, systemic fibrinolytic activity in dogs was assessed by ROTEM and standard plasma coagulation tests. A comparison was made between dogs with ascites and/or pleural effusions and control sick dogs without any type of intracavitary effusion. Group 1 included dogs with any type of abdominal and/or pleural effusion, which was confirmed by thoracic radiography, ultrasonography or computer tomography, and who presented to the San Marco Veterinary Clinic from March 2008 to November 2016. Intracavitary effusions in group 1 dogs were classified according to their pathophysiology of formation. Diagnosis of the disease causing the intracavitary effusion was used as the criterion to establish the pathophysiology of the fluid formation. Group 2 included sick dogs without any type of intracavitary effusion. They were chosen from the electronic medical database, P.O.A System-Plus 9.0$^{\circledR}$, and included dogs that presented to the San Marco Veterinary Clinic during the same period of time as those in group 1. All control dogs were individually matched to group 1 dogs type of underlying disease based on their. When 2 or more dogs fulfilled these criteria, the choice for the pairing was randomly made by the computer system P.O.A System-Plus 9.0$^{\circledR}$. To be included in the study, dogs of both groups were required to have received a specific diagnosis relevant to the initial reason for examination. They also had a complete medical record, including history and results of physical examination. Assays completed at the time of presentation were a complete blood count (including blood smear examination), serum biochemistry analysis, standard coagulation tests and ROTEM assay, and urinalysis. Dogs of both groups were excluded from the study if they had been treated with plasma, plasma derivates, anticoagulant therapy, tranexamic acid, or for anticoagulant intoxication within the 30 days prior to study enrolment.

Localization and type of effusion in dogs of groups 1, and causes of sickness for both groups of dogs were described. Gender, sexual status, age, pure breed *vs.* mongrel status, body weight and frequency of clinically relevant bleeding for all the dogs included in the 2 groups were recorded and used in comparisons. For the purpose of this study, clinically relevant bleeding was defined as any bleeding that caused anemia, defined as a hematocrit (HCT) $< 39.0\%$ (reference interval 39.0–59.2).

### Analysis of ROTEM and standard plasma coagulation tests

Venous blood samples for ROTEM point of care and plasma standard coagulation tests were taken from the cephalic (for medium/large-size dogs) or jugular (for small size-dogs) veins. In all dogs, venous blood samples for both ROTEM and standard plasma coagulation tests were taken at the same time (± 2 hours). Two 3.5 mL aliquots of blood were transferred in plastic tubes, containing 3.2% sodium citrate (3.2% sodium citrate Vacuette$^{\circledR}$ 3.5 mL, Grenier Bio-One, Kremsmünster, Austria) to give a final ratio of volumes of anticoagulant to blood of 1:9, for the measurement of both ROTEM and standard plasma coagulation tests. ROTEM parameters analyzed in this study included A5, LI60 and ML for a 2-hour runtime (ML/2h) in EXTEM assays. Plasma standard coagulation tests analyzed in this study included fibrinogen, semiquantitative FDPs, and D-dimer concentrations.

EXTEM assays on ROTEM (Pentapharm GmbH, Munich, Germany) were performed according to the manufacturer's recommendations. Analyses were started within 10 minutes of blood sampling. Briefly, according to the pipetting programme, 20 µL re-calcification reagent (0.2 mmol/L calcium chloride solution; StarTEM, TEM Innovations GmbH, Munich, Germany) and 20 µL EXTEM activation reagent (human recombinant tissue factor; EXTEM,

TEM Innovations GmbH, Munich, Germany) were added to a pre-warmed cup. Then, 300 μL whole blood containing 0.32% sodium citrate was added to the cup and, after a semi-automated mixing step, the cup holder was placed in the measuring position of the ROTEM device. The measurements, run at 38.0˚C, were stopped after 120 min according to the protocol. After measurement, all ROTEM tracings were visually evaluated for artefacts. Of the parameters measured and calculated by the ROTEM device, EXTEM A5, LI60 and ML/2h were further evaluated by statistical analysis.

Tubes with whole blood containing 0.32% sodium citrate were centrifuged at 1,950 *g* for 5 min, plasma was harvested, and standard coagulation tests analysis was performed within 1 hour of blood sample collection. Plasma fibrinogen concentrations were determined by quantitative assays, using STA Fibrinogen (Diagnostica Stago, Asnières sur Seine, France), with a STA-R Evolution automated analyzer (Diagnostica Stago, Roche, Bäsel, Switzerland). The detection limit for fibrinogen was 60 mg/dL; for statistical analysis values below this concentration were entered into the data sheet as 59 mg/dL. Plasma concentrations of FDPs were determined using a semiquantitative plasma latex agglutination kit, FDPs Plasma, (Diagnostica Stago, Asnières sur Seine, France) that was validated for use with canine blood [30,31]. Plasma D-dimer concentrations were determined using a validated [32], immuno-turbidimetric quantitative assay, Tina-quant D-dimer (Roche Diagnostic GmbH, Mannheim, Germany), with an Olympus AU 2700 automated analyzer (Olympus Diagnostics, Hamburg, Germany).

For all ROTEM and standard plasma coagulation tests, internal laboratory reference intervals were calculated from 40 and 120 clinically healthy randomly selected dogs, respectively.

All collection procedures were performed solely as part of the dog's prescribed health care and for standard diagnostic and monitoring purposes. Previous informed written consent was obtained from all dog owners. No anesthesia, euthanasia, or any kind of animal sacrifice was required in this study. All procedures complied with the European Union legislation "on the protection of animals used for scientific purposes" (Directive 2010/63/EU) and with the ethical requirements of Italian law (Decreto Legislativo 04/03/2014, n. 26). Accordingly, this study did not require authorization or an ID protocol number.

## Fibrinolysis definitions

Based on the ROTEM assay results and plasma fibrinogen concentration, fibrinolysis severity grading for this study was:

1. Hypofibrinolysis and basal fibrinolysis were LI60 values higher or within the internal laboratory reference interval of 86% - 98%, respectively.

2. Increased fibrinolysis was LI60 values smaller than the internal laboratory reference interval of 86% with plasma fibrinogen concentration > 100 mg/dL.

3. Hyperfibrinolysis was LI60 smaller than the internal laboratory reference interval of 86% with plasma fibrinogen concentration ≤ 100 mg/dL.

From the categorization introduced by Schöchl et al., for dogs with LI60 values smaller than the internal laboratory reference interval of 86%, 3 patterns of fibrinolysis were identified, based on the time course of clot breakdown: (a) fulminant with a total breakdown of the clot within 30 minutes, (b) intermediate with a total breakdown between 30 and 60 minutes, and (c) late with a breakdown of the clot after 60 minutes [33].

Based plasma semiquantitative FDPs, D-dimer, and plasma fibrinogen concentrations, fibrinolysis severity grading for this study was:

1. Hypofibrinolysis or basal fibrinolysis had normal FDPs and D-dimer concentrations.

2. Increased fibrinolysis had increased FDPs and/or D-dimer concentrations with fibrinogen concentration > 100 mg/dL. Increased fibrinolysis had further sub-types:

   a. Primary fibrinolysis had increased FDPs and normal D-dimer concentrations with fibrinogen concentrations > 100 mg/dL;

   b. Secondary fibrinolysis had normal to increased FDPs and increased D-dimer concentrations with fibrinogen concentrations > 100 mg/dL.

3. Hyperfibrinolysis had increased FDPs and/or D-dimer concentrations with fibrinogen concentration ≤ 100 mg/dL. Hyperfibrinolysis had further sub-types:

   a. Primary hyperfibrinolysis had increased FDPs and normal D-dimer concentrations with fibrinogen concentrations ≤ 100 mg/dL;

   b. Secondary hyperfibrinolysis had normal to increased FDPs and increased D-dimer concentrations with fibrinogen concentrations ≤ 100 mg/dL.

### Statistical analysis

Continuous data were assessed for normality of distribution with the Shapiro-Wilk test. Normally distributed data are reported as a mean ± standard deviation (SD), and non-normally distributed data were reported as a median and interquartile range (IQR).

Differences of sexual status ($\chi^2$-test), pure breed *vs* mongrel dogs (Fisher exact test), age (t-test), body weight (t-test), and clinically relevant bleeding ($\chi^2$-test with Yates correction) were evaluated between groups 1 and group 2.

EXTEM A5 (t-test), LI60 (Mann-Whitney test), ML/2h (Mann-Whitney test), plasma fibrinogen (Mann-Whitney test), semiquantitative FDPs ($\chi^2$-test), and D-dimer (Mann-Whitney test) concentrations were compared between group 1 and group 2.

Differences between group 1 and 2 in fibrinolysis grade of severity (hypofibrinolysis or basal fibrinolysis, increased fibrinolysis and hyperfibrinolysis) assessed by ROTEM assay results and plasma fibrinogen concentration and by plasma semiquantitative FDPs, D-dimer, and fibrinogen concentrations were evaluated by Fisher exact test. Agreement in fibrinolysis grade of severity assessed by these 2 methods was evaluated by Cohen's Kappa statistic.

Differences between group 1 and 2 in pattern of fibrinolysis (hypofibrinolysis or basal fibrinolysis, late fibrinolysis, intermediate fibrinolysis and fulminant fibrinolysis) assessed by ROTEM assay results were evaluated by Fisher exact tests.

Differences between group 1 and group 2 in type of fibrinolysis (hypofibrinolysis or basal fibrinolysis, primary fibrinolysis, secondary fibrinolysis, primary hyperfibrinolysis and secondary hyperfibrinolysis) assessed by plasma semiquantitative FDPs, D-dimer, and fibrinogen concentrations were evaluated by Fisher exact test.

Clinically relevant bleeding can cause hypoperfusion and the latter has been associated with hyperfibrinolysis in both humans and dogs [34,35]. Therefore, viscoelastic measures of fibrinolysis (i.e., EXTEM A5, LI60, and ML/2h), standard plasma coagulation tests (i.e., plasma fibrinogen, semiquantitative FDPs, and D-dimer concentrations), ROTEM fibrinolysis grade of severity, and ROTEM pattern of fibrinolysis were also evaluated excluding dogs with clinically relevant bleeding from both groups. Group 1 and 2 without clinically relevant bleeding dogs were named group 1A and 2A, respectively.

For all statistical analyses, the significance level was set to α = 0.05.

## Results

### Animals

During the study period, 1619 dogs with any type of abdominal and/or pleural effusion, as confirmed by thoracic radiography, ultrasonography or computer tomography, presented to the clinic. For 34 of these dogs, plasma samples for both standard coagulation tests and ROTEM assay were collected at presentation. Two dogs were excluded from further analysis, 1 due to rodenticide exposure and 1 due to tranexamic acid treatment. The remaining 32 dogs entered the study in group 1. Twenty-five of these dogs had only abdominal effusions, including 2 transudates due to decreased colloid osmotic pressure, 10 transudates due to increased hydrostatic pressure, 3 exudates, and 10 hemorrhagic effusions. Six dogs had concomitant abdominal and pleural effusions, including 5 transudates due to increased hydrostatic pressure and 1 hemorrhagic effusion in both the abdominal and the thoracic cavities. One dog had only a pleural effusion, an exudate. Dogs with intracavitary effusions (group 1) and sick dogs without any type of intracavitary effusion (group 2) were matched for type of underlying disease which included in each group: neoplasia (n = 15), cardiac problems (5), liver disease (3), inflammatory or immune-mediated disease (3), traumatic disease (3), gastrointestinal disorders (2), and sepsis and infectious disease (1).

Group 1 dogs included 17 males, 16 (50%) sexually intact and 1 (3.1%) neutered, and 15 females, 3 (9.4%) sexually intact and 12 (37.5%) spayed. Twenty (62.5%) dogs were pure breed and 12 (37.5%) were mongrels. Mean age was 107 ± 40 months and mean body weight was 26.7 ± 15.1 kg. Clinically relevant signs of bleeding were present in 13 of these dogs. One case was trauma-related, 11 secondary to rupture of pathologic/neoplastic organs, 1 secondary to rupture of pathologic/neoplastic organs and concurrent severe thrombocytopenia, and in 1 case secondary to idiopathic pericarditis.

Group 2 dogs included 14 males, 7 (21.9%) sexually intact and 7 (21.9%) neutered, and 18 females, 5 (15%) sexually intact and 13 (40%) spayed. Seventeen (53.1%) dogs were pure breed and 15 (46.9%) were mongrels. Mean age was 108 ± 47 months and mean body weight was 22.1 ± 14.7 kg. Clinically relevant signs of bleeding were present in 4 of these dogs. One case was secondary to trauma, 1 secondary to rupture of pathologic/neoplastic organs, 1 secondary to severe thrombocytopenia, and 1 secondary to surgery in a dog affected by von Willebrand disease.

There was a statistical difference regarding sexual status ($\chi^2$ = 8.56, $P$ = 0.036) with more intact males in group 1 compared to group 2. There was an overall 53% breed match between groups 1 and group 2 with no statistical difference found in the percentage of pure breed *vs* mongrel dogs ($P$ = 0.61), age (t = -0.07, $P$ = 0.94), and body weight (t = 1.23, $P$ = 0.22). Frequency of clinically relevant bleeding was significantly higher in dogs with intracavitary effusions compared to the controls ($\chi^2$ = 5.13, $P$ = 0.023).

### ROTEM assay and plasma fibrinogen, FDPs and D-dimer concentrations

EXTEM A5 (reference interval, 29–44 mm) was significantly smaller in group 1 (mean = 34.88 ± 16.23 mm) compared to group 2 (mean = 53.41 ± 16.24 mm; t = -4.57, $P$ < 0.001; Fig 1A). EXTEM LI60 (reference interval 86%– 98%) was significantly lower in group 1 (median = 95%, IQR 61% – 100%) compared to group 2 (median = 99%, IQR 97% – 99%; U = 366, $P$ = 0.047; Fig 1B). EXTEM ML/2h (reference interval 19%– 38%) was significantly higher in group 1 (median = 18.5%, IQR 5% – 47%) compared to group 2 (median = 6%, IQR 3% – 14%; U = 290, $P$ = 0.003; Fig 1C). Finally, also when dogs with clinically relevant bleeding were excluded from analysis EXTEM A5, LI60, and ML/2h, remained significantly smaller, lower and higher, respectively, in group 1A compared to group 2A (Table 1).

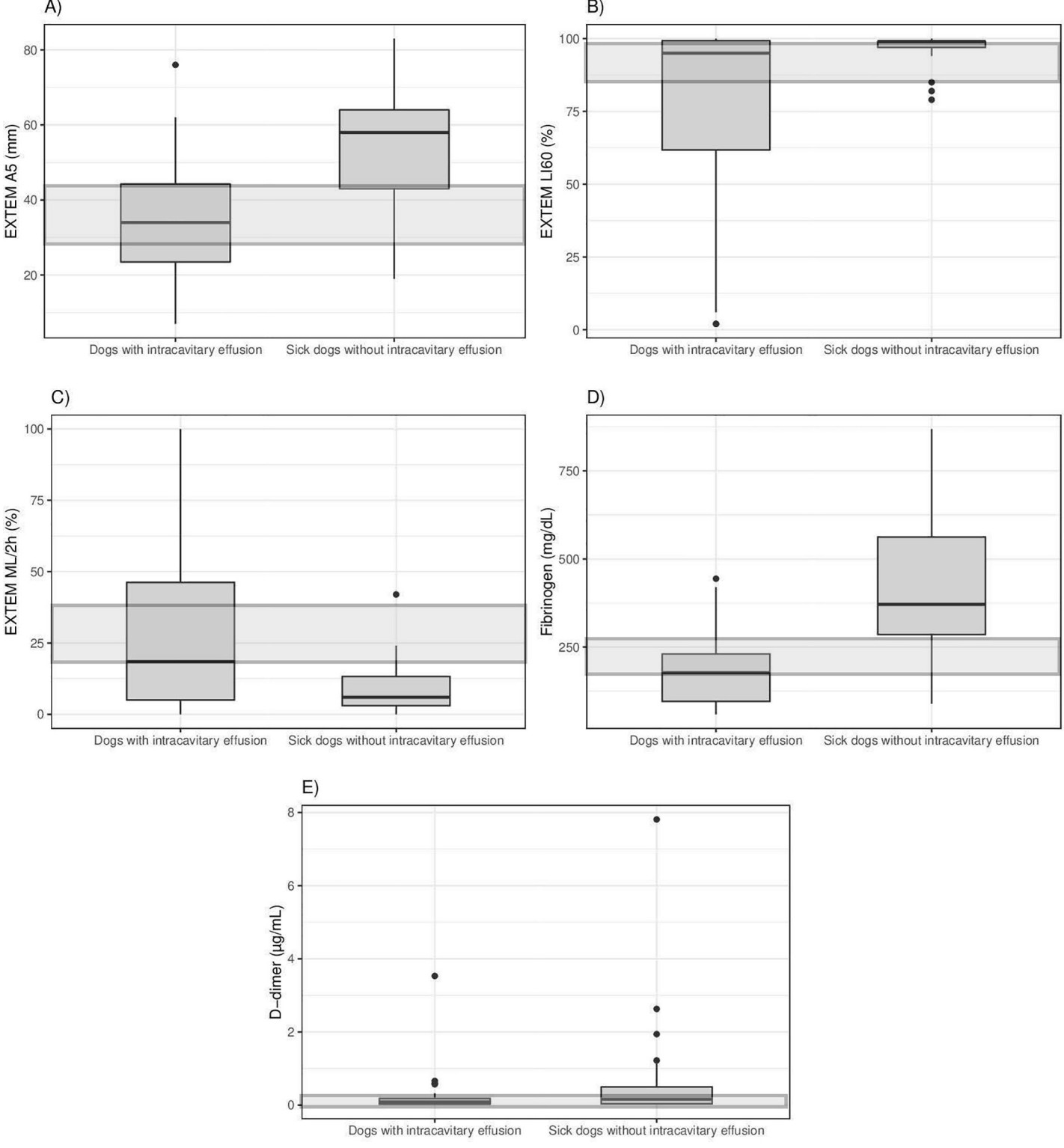

**Fig 1. Tukey boxplots of 3.2% citrated whole blood EXTEM A5 (A), 3.2% citrated whole blood EXTEM LI60 (B), 3.2% citrated whole blood EXTEM ML/2h (C), plasma fibrinogen concentrations (D), and plasma D-dimer concentrations (E) from dogs with intracavitary effusion (n = 32) and control sick dogs without intracavitary effusion (n = 32). A:** Data distributions of the two groups of dogs are significantly different ($P < 0.001$). **B:** Data distributions of the two groups of dogs are significantly different ($P = 0.047$). **C:** Data distributions of the two groups of dogs are significantly different ($P = 0.003$). **D:** Data distributions of the two groups of dogs are significantly different ($P < 0.001$). **E:** Data distributions of the two groups of dogs are not significantly different ($P = 0.262$). The shaded regions represent the

reference interval for each test. The bottom and top of the box are the 1$^{st}$ and 3$^{rd}$ quartiles; the median is the black line inside the box. Whiskers correspond to the lowest datum still within 1.5 IQR of the lower quartile, and the highest datum still within 1.5 IQR range of the upper quartile. Black dots are outlier values (> 1.5 IQR away from the closest end of the box). IQR: interquartile range.

Plasma fibrinogen concentrations (reference interval, 152–284 mg/dL) were significantly lower in group 1 (median = 177 mg/dL, IQR 96 – 234 mg/dL) compared to group 2 (median = 371 mg/dL, IQR 276 – 569 mg/dL; U = 144.5, P < 0.001; Fig 1D). No statistical difference was found in plasma concentrations of semiquantitative FDPs (reference interval < 5 µg/mL) between group 1 and group 2 ($\chi^2$ = 3.23, P = 0.199; Table 2). No statistical difference was also found in plasma D-dimer concentrations (reference interval 0.01–0.34 µg/mL) between group 1 (median = 0.09 µg/mL, IQR 0.02 – 0.2 µg/mL) and group 2 (median = 0.16 µg/mL, IQR 0.03 – 0.54 µg/mL; U = 429, P = 0.262; Fig 1E). Finally, also when dogs with clinically relevant bleeding were excluded from analysis plasma fibrinogen concentrations remained significantly lower in group 1A compared to group 2A, while no statistical difference was found in plasma concentrations of semiquantitative FDPs and D-dimer concentrations (Table 1).

## Fibrinolysis

There was a significant difference in fibrinolysis grade of severity (i.e., hypofibrinolysis or basal fibrinolysis, increased fibrinolysis, and hyperfibrinolysis) assessed by ROTEM assay results and plasma fibrinogen concentrations between group 1 and group 2 (P = 0.042), with group 1 dogs having a lesser degree of hypofibrinolysis and basal fibrinolysis and a higher degree of increased fibrinolysis and hyperfibrinolysis compared to dogs of group 2 (Table 3). This difference was still present when dogs with clinically relevant bleeding were excluded

**Table 1. ROTEM and standard plasma coagulation tests evaluated excluding dogs with clinically relevant bleeding.**

| Parameters | Group 1A (n = 19) | Group 1B (n = 28) | Test values | P-values |
|---|---|---|---|---|
| **EXTEM A5 (mm)** | | | | |
| mean ± SD | 40.47 ± 15.9 | 53.82 ± 16.47 | t = -2.76 | P = 0.008 |
| (RI = 29–44) | | | | |
| **EXTEM LI60 (%)** | | | | |
| median (IQR) | 87 (32 – 99) | 99 (97 – 96) | U = 402 | P = 0.003 |
| (RI = 86–98) | | | | |
| **EXTEM ML/2h (%)** | | | | |
| median (IQR) | 27 (9 – 80) | 7 (3 – 15) | U = 96 | P < 0.001 |
| (RI = 19–38) | | | | |
| **Fibrinogen (mg/dL)** | | | | |
| median (IQR) | 187 (95 – 248) | 373 (267 – 574) | U = 92 | P < 0.001 |
| (RI = 152–284) | | | | |
| **Semiquantitative** | <5, n = 7 | <5, n = 14 | | |
| **FDPs (µg/mL)** | 5–20, n = 8 | 5–20, n = 8 | $\chi^2$ = 1.04 | P = 0.592 |
| (RI < 5) | > 20, n = 4 | > 20, n = 6 | | |
| **D-dimer (µg/mL)** | | | | |
| median (IQR) | 0.8 (0.01 – 0.12) | 0.15 (0.01 – 0.43) | U = 209 | P = 0.222 |
| (RI = 0.01–0.34) | | | | |

Group 1A: dogs with intracavitary effusion without clinically relevant bleeding; group 2A: sick dogs without intracavitary effusions and without clinically relevant bleeding; IQR: interquartile range; RI: reference interval.

**Table 2. Plasma concentrations of semi quantitative FDPs in dogs with intracavitary effusions and control sick dogs without intracavitary effusions.**

| | FDPs (µg/mL) | | |
| --- | --- | --- | --- |
| | (RI <5) | | |
| | < 5 | ≥ 5 < 20 | ≥ 20 |
| **Group 1** | 9 (28.12%) | 11 (34.38%) | 12 (37.50%) |
| **(n = 32)** | | | |
| **Group 2** | 16 (50.0%) | 8 (25.0%) | 8 (25.0%) |
| **(n = 32)** | | | |

Data are the No. (%) of dogs. Group 1: dogs with intracavitary effusions; group 2: sick dogs without intracavitary effusions. RI: reference interval.

from analysis, with dogs of group 1A having a lesser degree of hypofibrinolysis and basal fibrinolysis and a higher degree of increased fibrinolysis and hyperfibrinolysis compared to dogs of group 2A ($P$ = 0.012; Table 3). ROTEM pattern of fibrinolysis (i.e., hypofibrinolysis or basal fibrinolysis, late fibrinolysis, intermediate fibrinolysis and fulminant fibrinolysis) was significantly different between group 1 and group 2 ($P$ = 0.044), with group 1 dogs having a lesser degree of hypofibrinolysis and basal fibrinolysis and a higher degree of late, intermediate, and fulminant fibrinolysis compared to dogs of group 2 (Table 4). This difference was still present when dogs with clinically relevant bleeding were excluded from analysis, with dogs of group 1A having a lesser degree of hypofibrinolysis and basal fibrinolysis and a higher degree of late, intermediate, and fulminant fibrinolysis compared to dogs of group 2A ($P$ = 0.009; Table 4).

There was also a significant difference in fibrinolysis grade of severity between group 1 and group 2 ($P$ = 0.017) when this was assessed by plasma semiquantitative FDPs, D-dimer, and fibrinogen concentrations, with group 1 dogs having a lesser degree of hypofibrinolysis and basal fibrinolysis and a higher degree of hyperfibrinolysis compared to dogs of group 2 (Table 5). This difference was still present when dogs with clinically relevant bleeding were excluded from analysis, with dogs of group 1A having a lesser degree of hypofibrinolysis and basal fibrinolysis and a higher degree of hyperfibrinolysis compared to dogs of group 2A ($P$ = 0.016; Table 5).

Within the 64 dogs included in the study and grouped together, the observed concordance in fibrinolysis grade of severity, assessed by ROTEM assay results and fibrinogen

**Table 3. Differences in fibrinolysis grade assessed by ROTEM in dogs with intracavitary effusions and control sick dogs without intracavitary effusions.**

| | Hypofibrinolysis | Basal fibrinolysis | Increased fibrinolysis | Hyperfibrinolysis |
| --- | --- | --- | --- | --- |
| **Group 1** | 12 (37.5%) | 9 (28.1%) | 5 (15.6%) | 6 (18.8%) |
| **(n = 32)** | | | | |
| **Group 2** | 17 (53.1%) | 12 (37.5%) | 2 (6.3%) | 1 (3.1%) |
| **(n = 32)** | | | | |
| **Group 1A** | 5 (26.3%) | 5 (26.3%) | 4 (21.1%) | 5 (26.3%) |
| **(n = 19)** | | | | |
| **Group 2A** | 15 (53.6%) | 10 (35.7%) | 2 (7.1%) | 1 (3.6%) |
| **(n = 28)** | | | | |

NB: for statistical analysis dogs with hypofibrinolysis and basal fibrinolysis were grouped together. Data are the No. (%) of dogs. Group 1: dogs with intracavitary effusions; group 2: sick dogs without intracavitary effusions; group 1A: dogs with intracavitary effusion and without clinically relevant bleeding; group 2A: sick dogs without intracavitary effusions and without clinically relevant bleeding.

**Table 4.** Differences in pattern of fibrinolysis assessed by ROTEM in dogs with intracavitary effusions and control sick dogs without intracavitary effusions.

| | Hypofibrinolysis | Basal fibrinolysis | Increased or hyperfibrinolysis | | |
| --- | --- | --- | --- | --- | --- |
| | | | Late | Intermediate | Fulminant |
| **Group 1** (n = 32) | 12 (37.5%) | 9 (28.1%) | 6 (18.8%) | 4 (12.5%) | 1 (3.1%) |
| **Group 2** (n = 32) | 17 (53.1%) | 12 (37.5%) | 3 (9.4%) | 0 (0%) | 0 (0%) |
| **Group 1A** (n = 19) | 5 (26.3%) | 5 (26.3%) | 5 (26.3%) | 3 (15.8%) | 1 (5.3%) |
| **Group 2A** (n = 28) | 15 (53.6%) | 10 (35.7%) | 3 (10.7) | 0 (0%) | 0 (0%) |

NB: for statistical analysis dogs with hypofibrinolysis and basal fibrinolysis were grouped together. Data are the No. (%) of dogs. Group 1: dogs with intracavitary effusions; group 2: sick dogs without intracavitary effusions; group 1A: dogs with intracavitary effusion and without clinically relevant bleeding; group 2A: sick dogs without intracavitary effusions and without clinically relevant bleeding.

concentrations and by plasma semiquantitative FDPs, D-dimer, and fibrinogen concentrations, was 34.4%. There was a poor agreement between the 2 classification schemes (K = 0.06, 95% CI = -0.14 – +0.26; Table 6).

There was a significant difference in type of fibrinolysis (i.e., hypofibrinolysis or basal fibrinolysis, primary fibrinolysis, secondary fibrinolysis, primary hyperfibrinolysis and secondary hyperfibrinolysis), assessed by plasma semiquantitative FDPs, D-dimer, and fibrinogen concentrations, between group 1 and group 2 (P = 0.004), with group 1 dogs having a higher frequency of primary fibrinolysis and primary hyperfibrinolysis and a lower frequency of secondary fibrinolysis compared to dogs of group 2 (Table 5). This difference was still present when dogs with clinically relevant bleeding were excluded from analysis, with dogs of group 1A still having a higher frequency of primary fibrinolysis and primary hyperfibrinolysis and a lower frequency of secondary fibrinolysis compared to dogs of group 2A (P = 0.004; Table 5).

**Table 5.** Differences in fibrinolysis grade of severity and type assessed by plasma semi quantitative FDPs, D-dimer, and fibrinogen concentrations in dogs with intracavitary effusions and control sick dogs without intracavitary effusions.

| | Hypofibrinolysis/ basal fibrinolysis | Increased fibrinolysis | | Hyperfibrinolysis | |
| --- | --- | --- | --- | --- | --- |
| | | PF | SF | PHF | SHF |
| **Group 1** (n = 32) | 8 (25.0%) | 15 (46.9%) | 2 (6.25%) | 5 (15.6%) | 2 (6.25%) |
| **Group 2** (n = 32) | 12 (37.5%) | 10 (31.25%) | 10 (31.25%) | 0 (0%) | 0 (0%) |
| **Group 1A** (n = 19) | 6 (31.6%) | 8 (42.1%) | 0 (0%) | 4 (21.0%) | 1 (5.3%) |
| **Group 2A** (n = 28) | 11 (39.3%) | 9 (32.1%) | 8 (28.6%) | 0 (0%) | 0(0%) |

NB: for statistical analysis dogs with PF and SF and dogs with PHF and SHF were grouped together only to assess differences in fibrinolysis grade of severity between groups 1 and 2 and between groups 1A and 2A. To assess type of fibrinolysis no subgroups were assembled together. Data are the No. (%) of dogs. Group 1: dogs with intracavitary effusions; group 2: sick dogs without intracavitary effusions; group 1A: dogs with intracavitary effusion and without clinically relevant bleeding; group 2A: sick dogs without intracavitary effusions and without clinically relevant bleeding. PHF: primary hyperfibrinolysis; PF: primary fibrinolysis, SHF secondary hyperfibrinolysis; SF: secondary fibrinolysis.

**Table 6. Observed concordance in fibrinolysis grade of severity assessed by ROTEM assay results and fibrinogen concentrations and by plasma semi quantitative FDPs, D-dimer results and plasma fibrinogen concentrations in the 64 dogs included in the study.**

| Fibrinolysis grade of severity assessed by | | ROTEM assay results and fibrinogen concentration | | |
|---|---|---|---|---|
| | | Hypofibrinolysis / basal fibrinolysis | Increased fibrinolysis | Hyperfibrinolysis |
| **Plasma semi quantitative FDPs, D-dimer and fibrinogen concentrations** | **Hypofibrinolysis / basal fibrinolysis** | 14 | 3 | 3 |
| | **Increased fibrinolysis** | 33 | 4 | 0 |
| | **Hyperfibrinolysis** | 3 | 0 | 4 |

## Discussion

The primary aim of this study was to assess if ROTEM could detect enhanced fibrinolytic activity in dogs with intracavitary effusions. The EXTEM A5, LI60, and ML/2h results supported the presence of an enhanced systemic fibrinolytic state in these dogs, as already detected by the use of the combination of plasma FDPs, D-dimers and fibrinogen concentrations in our previous studies [11–13]. The enhanced systemic fibrinolytic activity in dogs with intracavitary effusion was still preset also when animals with clinically relevant bleeding were excluded from analysis from both groups, suggesting that this result was not exclusively driven by a possible hypoperfusion due to hemorrhage. EXTEM A5 correlates with platelet count and fibrinogen concentration [36], and can be used for early detection of fibrinolysis [28]. An EXTEM A5 ≤ 35 mm can identify more than 90% of human patients developing hyperfibrinolysis [29]. Our findings of a smaller EXTEM A5 and a lower fibrinogen concentration in dogs with intracavitary effusion would therefore be in agreement with these reports. Moreover, the lower EXTEM LI60 and the higher EXTEM ML/2h found in these dogs would suggest that the smaller EXTEM A5 and the lower fibrinogen concentration were, at least in part, due to an enhanced fibrinolytic state, also making EXTEM A5 an early marker of fibrinolysis in dogs.

Schöchl et al. [33], and other authors [37–39], arbitrarily used the term 'hyperfibrinolysis' for lysis greater than a certain maximal amplitude on TEG/ROTEM testing (Schöchl uses EXTEM ML/1h > 15%). However, confusion has arisen with this viscoelastometric-associated terminology because traditionally hyperfibrinolysis describes a situation in which fibrinolytic activity is greater than fibrin formation, clot integrity is threatened, and there is clot breakdown [40], rather than a loose term used simply to describe increased evidence of fibrinolysis [38,41–44]. Therefore, the term 'TEG/ROTEM hyperfibrinolysis' has been suggested in relation to the viscoelastometric measurements [40]. In an attempt to combine the traditional hyperfibrinolysis definition with the viscoelastometric-associated terminology, and for the purpose of this study, we decided to define hyperfibrinolytic a ROTEM trace only when the EXTEM LI60 exceeded the reference interval and the fibrinogen concentration was ≤ 100 mg/dL. If the fibrinogen concentration was > 100 mg/dL, it was defined as "increased fibrinolysis". The rationality was to align the definition of viscoelastometric hyperfibrinolysis, based on increased evidence of fibrinolysis at 60 min [33,41,43,44], with heighted risk of bleeding, as suggested by the traditional hyperfibrinolysis definition. A fibrinogen concentration ≤ 100 mg/dL was chosen as an index of "increased risk of bleeding", as fibrinogen concentrations below this value were associated with a high annual percentage of spontaneous bleeding in human patients with afibrinogenemia or hypofibrinogenemia [45]. The fibrinogen cutoff concentration was based on human data due to the lack of similar studies in dogs and the comparable physiological fibrinogen concentrations in these 2 species [46]. Using our ROTEM definition of increased fibrinolysis and hyperfibrinolysis, we demonstrated that dogs presenting with intracavitary effusions not only had a smaller EXTEM A5 and LI60 with higher ML/

2h, but also a more severe grade of fibrinolysis compared to the controls. Again, this result did not change when animals with clinically relevant bleeding were excluded from analysis from both groups. Because these 2 groups of dogs were matched for type of underlying disease, it is possible, that the higher grade of fibrinolysis observed in these dogs was due to the presence of intracavitary fluid, as previously suggested in both humans and dogs [11–15]. However, our study was not designed to demonstrate that the presence of an intracavitary effusion was the cause of this result. In fact, the presence of the intracavitary effusion could be just a marker identifying dogs with a more severe disease, with the severity being the cause of the enhanced fibrinolysis. This could be a possibility because, in our matching strategy for type of underlying disease, we have not matched the dogs also by disease severity due to the lack of a scoring system applicable to all the included illnesses. However, in a previous study we demonstrated that, at least in dogs with congestive heart failure, it was not the severity of the cardiac disease but the presence of ascites that was associated with, and therefore the possible cause, of hyperfibrinolysis [11].

One advantage of studying fibrinolysis by ROTEM assay compared to FDPs and D-dimer concentrations is that it allows to recognize 3 patterns of fibrinolysis [33]. Our results showed that in dogs with intracavitary effusion late, intermediate and fulminant fibrinolysis were present in 18.8%, 12.5% and 3.1% of the cases, respectively. By contrast, only 9.4% of the control dogs had a late fibrinolysis, while none of them had intermediate or fulminant fibrinolysis. These results clearly showed that dogs with intracavitary effusion had an increased frequency and a more severe ROTEM pattern of fibrinolysis compared to dogs without intracavitary effusions. Again, this result was confirmed also when animals with clinically relevant bleeding were excluded from analysis from both groups. Although this study, and our previous studies [11–13], found that clinically relevant bleeding was more frequent in dogs with intracavitary effusion, most hemorrhage resulted from rupture of pathological/neoplastic organs or secondary to trauma. The lack of spontaneous bleeding, despite enhanced fibrinolysis, may be explained by the low frequency of fulminant fibrinolysis, which is the ROTEM pattern of fibrinolysis associated with the worse outcome in humans [33,47]. Moreover, when fulminant fibrinolysis is present, bleeding may not occur in the absence of trauma or rupture of a pathological organs [48], even if the hyperfibrinolysis lead to severe hypofibrinogenemia [49]. Due to the retrospective nature of this study, it was not possible to establish if the increased/hyper fibrinolysis preceded or followed the bleeding event; nevertheless, from a biological point of view it was likely that hypotension and accumulation of intracavitary fluid, which both followed the bleeding episode, triggered the fibrinolysis [11–15,29,33,34,50]. It is certainly possible that the increased/hyper fibrinolysis could have then contributed to the magnitude of the bleeding events, threatening clots formation and stability. Beyond the increased/hyper fibrinolysis as a possible cause for the higher frequency of clinically relevant bleeding in dogs with intracavitary effusion, the design of our study might also have played a part in this result. In fact, by default, all dogs with intracavitary hemorrhage were included in the studied group and not in the controls.

A second advantage of studying fibrinolysis by ROTEM assay compared to obtaining FDPs and D-dimer concentrations is that it allows dogs with hypofibrinolysis to be distinguished from those with basal fibrinolysis. Although an investigation of hypofibrinolysis is beyond the scope of this study, it should be noted that, at least in humans, hypofibrinolysis seems to play a major role in the pathophysiology of myocardial infarction, thrombosis, sepsis, and disseminated intravascular coagulation [28], and that the presence of a hypofibrinolytic state in some clinical conditions has been associated with poorer outcomes [37–39]. Therefore, its identification may be important despite viscoelastometric tests have been considered unsuitable for the detection of hypofibrinolysis by some authors [51].

The secondary aim of this study was to describe the grade of severity of fibrinolysis detected by ROTEM and by FDPs, D-dimer and fibrinogen concentrations, and to assess the degree of agreement between these 2 classification schemes. In a manner similar to how we aligned the traditional hyperfibrinolysis definition with viscoelastometric-associated terminology, we also graded fibrinolysis as increased fibrinolysis or hyperfibrinolysis based on a combination of the semiquantitative FDPs, D-dimer and fibrinogen concentrations. A fibrinogen concentration equal/below or above 100 mg/dL, respectively, was set as the cut-off values for the 2 grades of fibrinolysis. Similarly to the ROTEM findings, also the use of FDPs and D-dimer showed that dogs with intracavitary effusions have a lesser degree of hypofibrinolysis and basal fibrinolysis, and a higher degree of hyperfibrinolysis compared to controls, and that this result was still present excluding dogs with clinically relevant bleeding from statistical analysis. Nevertheless, the overall agreement between the 2 classification schemes was poor, with most disagreement being caused by dogs classified as hypofibrinolytic or with basal fibrinolysis by ROTEM as having an increased fibrinolysis when classified by FDPs and D-dimer concentrations (Table 6). A possible explanation for this discrepancy is that while ROTEM assay can detect severe fibrinolysis, it is not sensitive enough to detect less severe forms of fibrinolysis [52–54]. Alternatively, plasma semiquantitative FDPs concentrations might have potentially overestimated the real grade of fibrinolysis present in our dogs. In fact, dogs with increased fibrinolysis were so classified due to their increased concentrations in FDPs (Table 5). Because this was the only semiquantitative and manually performed test in our study, it is possible that the use of a more robust quantitative and automated FDPs assay could have led to different results. A final hypothesis for the discrepancy between ROTEM assays and FDPs and D-dimer concentrations is that while ROTEM detects increased/hyper fibrinolysis as long as the production of plasmin within the general circulation overwhelms the neutralizing capacity of the antiplasmin, the FDPs and D-dimer concentrations suggest the presence of an increased/hyper fibrinolysis after its onset and beyond its duration, due to the time needed for the production of these products and their subsequent clearance to basal concentrations [48]. Euglobulin lysis time is a test of global fibrinolysis, and it has been used for estimating the functional fibrinolytic capacity of plasma and as gold standard in assessing fibrinolysis [41,55,56]. However, in the absence of this test, or any other third test for evaluating fibrinolysis (e.g. plasmin-α2-antiplasmin complex), it was not possible to determine whether the ROTEM assay or the FDPs and D-dimer concentrations misclassified the real grade of fibrinolysis severity in our dogs.

One advantage of studying fibrinolysis by FDPs, D-dimer and fibrinogen concentrations compared to ROTEM assay is that it allows to distinguish primary versus secondary increased/ hyper fibrinolysis. This distinction, which cannot be extrapolated from ROTEM assay results, is crucial for decisions on antifibrinolytic treatment in patients with clinically relevant hyperfibrinolysis [4]. At the moment rapid evaluation of the current status of the fibrinolytic system remains a challenge, making decision on antifibrinolytic treatment, for example in human trauma patients where degree and type of fibrinolysis are very variable, still empiric [57]. In our study, while there was a poor (i.e., 34.4%) overall agreement between ROTEM and plasma semiquantitative FDPs, D-dimer and fibrinogen concentrations in assessing grade of severity of fibrinolysis, this percentage rose to 40% when we only considered cases of possible hyperfibrinolysis (Table 6). Therefore, it might be suggested to bypass the empiric use of tranexamic acid in patient with fibrinolysis, performing in parallel both ROTEM and plasma determination of semiquantitative FDPs, D-dimer and fibrinogen. If both classification methods indicated the presence of hyperfibrinolysis, then semiquantitative FDP and D-dimer concentrations could be used in guiding antifibrinolytic therapy only for the cases of primary hyperfibrinolysis.

There are other 2 limitations to our study. First, in attempting to study the relevance of intracavitary effusion on the fibrinolytic system, we matched dogs in the study group (group

1) with those in a control population (group 2), based on type of disease. In doing so, we end up with a statistically significant increased number of sexually intact males with intracavitary effusions compared to the controls. It was possible that the variations in testosterone concentrations between the 2 groups could have influenced the results. Decreases in testosterone concentrations in humans are associated with an enhancement of fibrinolytic inhibition via increased synthesis of the plasminogen activator inhibitor-1 [58]. However, in all our previous studies we found enhanced fibrinolytic activity in dogs with intracavitary effusions, despite the control populations being 100% matched for gender and sexual status [11–13]. Second, none of the dogs in our study population had variables in extrinsically activated assays without aprotinin (EXTEM) compared with corresponding variables from extrinsically activated thromboelastometric assays with aprotinin (APTEM, TEM Innovations GmbH, Munich, Germany), as suggested for early diagnosis of fibrinolysis [41,59–61]. Nevertheless, a recent study in humans found that fibrinolysis seen in EXTEM tracings was a reliable finding and that APTEM assays failed to improve the accuracy of the diagnosis [29].

## Conclusions

In summary, ROTEM assay results supported our previous findings suggesting the presence of an enhanced fibrinolytic state in dogs with intracavitary effusions [11–13]. Moreover, dogs with intracavitary effusions showed an increased frequency, and a different and more severe pattern of fibrinolysis, compared to dogs without intracavitary effusions. An enhanced fibrinolytic state in dogs with intracavitary effusion was also detected by determining plasma semi-quantitative FDPs, D-dimer and fibrinogen concentrations. The overall agreement between the 2 classification schemes was poor. Further studies need to be carried out to determine if concurrent use of the 2 classification methods in selected cases might aid decision making on antifibrinolytic agent administration in patient with hyperfibrinolysis.

## Supporting information

**S1 File. Minimal anonymized data set.** Excel dataset of the patients' clinical records included in the present study.
(XLSX)

## Author Contributions

**Conceptualization:** Andrea Zoia, Michele Drigo, Christine J. Piek, Helena Calcini, Marco Caldin, Paolo Simioni.

**Data curation:** Andrea Zoia, Michele Drigo, Helena Calcini.

**Formal analysis:** Andrea Zoia, Michele Drigo, Helena Calcini, Marco Caldin.

**Funding acquisition:** Christine J. Piek, Marco Caldin.

**Investigation:** Andrea Zoia, Michele Drigo, Christine J. Piek, Helena Calcini, Marco Caldin, Paolo Simioni.

**Methodology:** Andrea Zoia, Michele Drigo, Christine J. Piek, Helena Calcini, Marco Caldin, Paolo Simioni.

**Project administration:** Andrea Zoia, Christine J. Piek, Marco Caldin.

**Resources:** Andrea Zoia, Marco Caldin.

**Supervision:** Michele Drigo, Christine J. Piek, Paolo Simioni.

**Writing – original draft:** Andrea Zoia, Michele Drigo, Christine J. Piek, Helena Calcini, Marco Caldin, Paolo Simioni.

**Writing – review & editing:** Andrea Zoia, Michele Drigo, Christine J. Piek, Helena Calcini, Marco Caldin, Paolo Simioni.

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
