## [Decision Letter · Decision Letter 0]

25 Jul 2019

PONE-D-19-15389

Enhanced fibrinolysis detection in a natural occurring canine model with intracavitary effusions: comparison and degree of agreement between thromboelastometry and FDPs, D-dimer and fibrinogen concentrations

PLOS ONE

Dear Mr Zoia,

Thank you for submitting your manuscript to PLOS ONE. After careful consideration, we feel that it has merit but does not fully meet PLOS ONE’s publication criteria as it currently stands. Therefore, we invite you to submit a revised version of the manuscript that addresses the points raised during the review process.

The authors should include healthy values for comparison obtained in the same conditions. Further, the presence of bleeding in both groups of studied animals makes difficult to interpret the results of the ROTEM analysis. 

The presentation of the results in the figures should be optimized. The graphs could be combined in one figure with different panels. Note that there is a typo in figure one ("intracavitay").

We would appreciate receiving your revised manuscript by Sep 08 2019 11:59PM. To enhance the reproducibility of your results, we recommend that if applicable you deposit your laboratory protocols in protocols.io, where a protocol can be assigned its own identifier (DOI) such that it can be cited independently in the future. For instructions see: http://journals.plos.org/plosone/s/submission-guidelines#loc-laboratory-protocols

We look forward to receiving your revised manuscript.

Kind regards,

Pablo Garcia de Frutos

Academic Editor

PLOS ONE

2. We noticed you have some minor occurrence of overlapping text with the following previous publications, which needs to be addressed:

-'Hemostatic findings of pleural fluid in dogs and the association between pleural effusions and primary hyperfibrino(geno)lysis: A cohort study of 99 dogs', https://doi.org/10.1371/journal.pone.0192371

-'Evaluation of rotation thrombelastography for the diagnosis of hyperfibrinolysis in trauma patients', https://doi.org/10.1093/bja/aen083

In your revision ensure you cite all your sources (including your own works), and quote or rephrase any duplicated text outside the methods section. Further consideration is dependent on these concerns being addressed.

3. Thank you for stating the following in the Competing Interests/Financial Disclosure* (delete as necessary) section:

"The author(s) received no specific funding for this work"

We note that one or more of the authors are employed by a commercial company: San Marco Veterinary Clinic, Padua, Italy.

Reviewers' comments:

Reviewer's Responses to Questions

**Comments to the Author**

1. Is the manuscript technically sound, and do the data support the conclusions?

Reviewer #1: Yes

Reviewer #2: Partly

2. Has the statistical analysis been performed appropriately and rigorously? 

Reviewer #1: No

Reviewer #2: N/A

3. Have the authors made all data underlying the findings in their manuscript fully available?

Reviewer #1: Yes

Reviewer #2: Yes

4. Is the manuscript presented in an intelligible fashion and written in standard English?

Reviewer #1: Yes

Reviewer #2: Yes

5. Review Comments to the Author

Reviewer #1: This is a well written manuscript with well described methods and statistics. My major concern about this study is the heterogenous patient population, which I am concerned limits the value of the data analysis.

The basic premise of the study is that patients with peritoneal and pleural effusions are more likely to have increased fibrinolysis than patients without effusions, but the authors present no rationale for this hypothesis. While they cite their previous work documenting traditional coagulation and fibrinolysis assays that support the concept, I am perplexed by the mechanisms that would underly this finding, and the authors do not attempt to explain why a dog with right sided heart failure would develop hyperfibrinolysis. Given that 10/23 dogs with effusions had hemoperitoneum, I am concerned that the findings in this study are driven largely by the dogs that bled. A statistical analysis (or at least descriptive statistics) showing how many of the dogs that demonstrated hyperfibrinolysis were hemoperitoneum cases and how many were not bleeding should be done to determine if all of these findings are driven by the bleeding dogs. I would argue that combining all of these types of disease processes into a single group makes these findings very difficult to interpret. Also having no data on the severity of the clinical signs and degree of shock in the bleeding patients makes it difficult to decide how to use the results of this study since pervious work has shown that the severity of shock is closely associated with viscoelastic measures of hyperfibrinolysis in dogs (Fletcher et al., JVECC, 2016). I worry that the small number of cases in this study and the heterogeneity of the group make the findings difficult to interpret and impossible to generalize. Without some sort of statistical analysis demonstrating that the hemoperitoneum cases are not solely driving these findings, I cannot endorse the conclusions of the authors.

The classification scheme proposed to categorize fibrinolysis in these patients is interesting and potentially useful, but I have some concerns that with the very small number of dogs in this study, these numerous classifications are confusing and of questionable benefit in the study. The arbitrary cutoffs for the ROTEM and traditional parameters evaluated would need to be validated in a larger study, and I worry that this complex scheme detracts from the findings of the paper. I would suggest that using a more simplified scheme denoting patients as hyperfibrinolytic, with normal systemic fibrinolysis, and hypofibrinolytic would improve the clarity of this paper.

Reviewer #2: In this veterinary clinical study, the authors select two groups of animals from a large cohort considering the presence of cavitary effusion. Their aim is to study the fibrinolytic status of these animals in relation to the presence of pleural, peritoneal, or pericardial effusions by ROTEM technology. The study has several limitations that should be considered by the authors. They do not have a healthy control study group to compare their results on sick animals. The authors label their control group as "Sick dogs without intracavitary effusion". However, in both groups the animals are sick. It would be important to have an adequate healthy group for comparison. The presence of bleeding in a portion of the individuals in both groups complicates the interpretation. The authors should include an analysis of their results if these animals are excluded.

6. PLOS authors have the option to publish the peer review history of their article (what does this mean?). If published, this will include your full peer review and any attached files.

Reviewer #1: No

Reviewer #2: No

---

## [Author Response · Author response to Decision Letter 0]

29 Aug 2019

RE: Manuscript PONE-D-19-15389

Enhanced fibrinolysis detection in a natural occurring canine model with intracavitary effusions: comparison and degree of agreement between thromboelastometry and FDPs, D-dimer and fibrinogen concentrations

Dear Dr. Pablo Garcia de Frutos, 

The authors would like to thank You and the Reviewers for the time spent and for their constructive comments in attempt to improve our manuscript. 

A number of changes (listed below) have been made to the current manuscript to incorporate the suggestions you made and to address the comments of the Reviewers. We hope that these are satisfactory to allow for publication. Belows follow our reply to your 3 specific requests.

1) The authors should include healthy values for comparison obtained in the same conditions. 

Authors’ reply: To give an idea for comparison to healthy values, reference intervals (obtained from healthy animals) for all the parameters studied were included (and still are) in the result section of our paper. On the other hand, if the Editor meant that we should include a healthy control population, as requested from Reviewer #2, we cannot adhere to this request because it would heavily damage the validity of our study (please see below). 

Case–control is a type of epidemiological observational study where subjects are observed in order to determine both their exposure and outcome status. A common misconception among investigators is that case-control design is perceived as a comparison between a group of diseased individuals versus a group of “normal” (i.e., disease-free) individuals. This conceptualization of the case-control design is inconsistent with the actual goal of control group selection, which is to provide a representative sample of the risk factors distribution among the population at risk. Therefore, following the current epidemiology textbooks indications and international guidelines on observational studies, we have included as a control population a group of sick dogs as much as possible similar to the specific diseased group (cases), but without the specific outcome of interest that we wanted to study (i.e., the presence of intracavitary effusion). On the other hand, a control group of healthy animals would have weakened the study, because different for too many aspects from a diseased group and not even representative of the general population. Selection of healthy subjects has been heavily criticized in the modern literature and the so-called “healthy patient bias” has been coined (REFERENCES follow). 

1) Bloom MS, Schisterman EF, Hediger ML. Selecting controls is not selecting “normals”: design and analysis issues for studying the etiology of polycystic ovary syndrome. Fertil Steril. 2006;86(1):2-12

2) Lewallen S, Courtright P. Epidemiology in practice: case-control studies. Comm Eye Health 1998;11:57-58

3) Wacholder JK, Silverman JS, McLaughlin JK, Mandel JS. Selection of controls in case-control studies. II. Types of Controls. Am J Epidemiol; 1992, 135:1029-1041

4) Shrank WH, Patrick AR, Brookhart A. Healthy user and related biases in observational studies of preventive interventions: a primer for physicians. J Gen Intern Med, 2011;26(5):546-550

2) Further, the presence of bleeding in both groups of studied animals makes difficult to interpret the results of the ROTEM analysis.

Authors’ reply: To address your concern additional statistical analysis excluding bleedings dogs from both groups were performed as requested. The additional analyses showed that bleeding is not the solely driving of our finding. In fact, excluding these dogs none of the results changed. Please see the 'Revised Manuscript with Track Changes’ at: 

• Materials and methods lines: 214-220

• Results: lines 269-271; 279-283; 357-360; 364-367; 372-374; Table 1, 3, 4, and 5.

• Discussion: 427–430; 462–462; 484–485; 520–521. 

3) The presentation of the results in the figures should be optimized. The graphs could be combined in one figure with different panels. Note that there is a typo in figure one ("intracavitay").

Authors’ reply: Following your instruction the 5 figures have been combined in one figure with 5 panels. The typo mistake has been corrected. 

Kind regards,

Andrea Zoia, DVM, Cert SAM, Dip ECVIM-CA

(Corresponding Author)

PS: If you have any further questions or comments, please do not hesitate to contact me.

Authors’ reply: following the above PDFs instructions, numerous changes have been made to the article’s format. If we still do not meet Plos One’s style requirements we would be grateful to receive more specific suggestions.

2. We noticed you have some minor occurrence of overlapping text with the following previous publications, which needs to be addressed:

-'Hemostatic findings of pleural fluid in dogs and the association between pleural effusions and primary hyperfibrino(geno)lysis: A cohort study of 99 dogs', https://doi.org/10.1371/journal.pone.0192371

-'Evaluation of rotation thrombelastography for the diagnosis of hyperfibrinolysis in trauma patients', https://doi.org/10.1093/bja/aen083

In your revision ensure you cite all your sources (including your own works), and quote or rephrase any duplicated text outside the methods section. Further consideration is dependent on these concerns being addressed.

Authors’ reply: Overlapping test in figure caption of this paper with our previous work “'Hemostatic findings of pleural fluid in dogs and the association between pleural effusions and primary hyperfibrino(geno)lysis: A cohort study of 99 dogs” was present and it has been drastically reduced/eliminated. There does remain some overlap between this article and the above article only for the author affiliation and for the references, which from our understanding should not constitute an issue. 

Small overlapping test in the introduction of this paper with the 'Evaluation of rotation thrombelastography for the diagnosis of hyperfibrinolysis in trauma patients' had been rephrased as requested. 

3. Thank you for stating the following in the Competing Interests/Financial Disclosure* (delete as necessary) section:

"The author(s) received no specific funding for this work"

We note that one or more of the authors are employed by a commercial company: San Marco Veterinary Clinic, Padua, Italy.

Authors’ reply: The San Marco Veterinary Clinic (Padua, Italy) did not play any role in the study nor altered our adherence to all PLOS ONE policies on sharing data and materials. An updated Funding Statement and a Competing Interests Statement has been included in the cover letter, as requested.

Reviewers' comments:

Reviewer's Responses to Questions

Comments to the Author

1. Is the manuscript technically sound, and do the data support the conclusions?

Reviewer #1: Yes

Authors’ reply: none requested

Reviewer #2: Partly

Authors’ reply: please see our reply to your comments below.

2. Has the statistical analysis been performed appropriately and rigorously? 

Reviewer #1: No

Authors’ reply: additional statistical analysis excluding dogs with bleeding has been added as requested (see our reply to your comments below). 

Reviewer #2: N/A

Authors’ reply: none requested

3. Have the authors made all data underlying the findings in their manuscript fully available?

Reviewer #1: Yes

Authors’ reply: none requested

Reviewer #2: Yes

Authors’ reply: none requested

4. Is the manuscript presented in an intelligible fashion and written in standard English?

Reviewer #1: Yes

Authors’ reply: none requested

Reviewer #2: Yes

Authors’ reply: none requested

5. Review Comments to the Author

Reviewer #1: 

1) This is a well written manuscript with well described methods and statistics. My major concern about this study is the heterogenous patient population, which I am concerned limits the value of the data analysis.

Authors’ reply: We thank the Reviewer to find our manuscript well written and with appropriate description on methods and statistics. To address your concern on the heterogenous patient population further statistical analysis excluding bleedings dogs from both groups were performed. For details see our reply to your point number 3. 

2) The basic premise of the study is that patients with peritoneal and pleural effusions are more likely to have increased fibrinolysis than patients without effusions, but the authors present no rationale for this hypothesis. While they cite their previous works documenting traditional coagulation and fibrinolysis assays that support the concept, I am perplexed by the mechanisms that would underlay this finding, and the authors do not attempt to explain why a dog with right sided heart failure would develop hyperfibrinolysis.

Authors’ reply: We thank the Reviewer for this comment which allowed us to improve our article. In our previous works cited in the current article there is well explained mechanism that would justify why dogs with peritoneal and pleural effusions are more likely to have increased fibrinolysis than patients without effusions, including dogs with right sided heart failure. It is true that this information is not present in the current article. Therefore, we briefly summarized rationale for this hypothesis also in this article following your comment, leaving the references to acquire more in-depth details for the readers more interested in such pathophysiological information. Moreover further 12 references (references 35 to 46) have been added to support this hypothesis. Please see 'Revised Manuscript with Track Changes’ at lines 465-468. 

3) Given that 10/23 dogs with effusions had hemoperitoneum, I am concerned that the findings in this study are driven largely by the dogs that bled. A statistical analysis (or at least descriptive statistics) showing how many of the dogs that demonstrated hyperfibrinolysis were hemoperitoneum cases and how many were not bleeding should be done to determine if all of these findings are driven by the bleeding dogs. I would argue that combining all of these types of disease processes into a single group makes these findings very difficult to interpret. Also having no data on the severity of the clinical signs and degree of shock in the bleeding patients makes it difficult to decide how to use the results of this study since pervious work has shown that the severity of shock is closely associated with viscoelastic measures of hyperfibrinolysis in dogs (Fletcher et al., JVECC, 2016). I worry that the small number of cases in this study and the heterogeneity of the group make the findings difficult to interpret and impossible to generalize. Without some sort of statistical analysis demonstrating that the hemoperitoneum cases are not solely driving these findings, I cannot endorse the conclusions of the authors.

Authors’ reply: We thank the Reviewer for this comment that allow us to improve our article. To address your concern additional statistical analysis excluding bleedings dogs from both groups were performed as requested. The additional analyses showed that bleeding is not the solely driving of our finding. In fact, excluding these dogs none of the results changed. Please see 'Revised Manuscript with Track Changes’ at: 

• Materials and methods lines: 214-220

• Results: lines 269-271; 279-283; 357-360; 364-367; 372-374; Table 1, 3, 4, and 5.

• Discussion: 427–430; 462–462; 484–485; 520–521. 

4) The classification scheme proposed to categorize fibrinolysis in these patients is interesting and potentially useful, but I have some concerns that with the very small number of dogs in this study, these numerous classifications are confusing and of questionable benefit in the study. The arbitrary cutoffs for the ROTEM and traditional parameters evaluated would need to be validated in a larger study, and I worry that this complex scheme detracts from the findings of the paper. I would suggest that using a more simplified scheme denoting patients as hyperfibrinolytic, with normal systemic fibrinolysis, and hypofibrinolytic would improve the clarity of this paper.

Authors’ reply: We thank the Reviewer to find the classification scheme proposed to categorize fibrinolysis in our patients interesting and potentially useful. Nevertheless, we realize that you have some concern with the very small number of dogs in this study being analysed in multiple subgroups. However, while in the descriptive statistics we have several groups (hypofibrinolysis, basal fibrinolysis, increased fibrinolysis, and hyerfibrinolys, with the latter two groups sometimes dived in primary and secondary) in our inferential statistics we used, to simplify the classification system, only 3 groups of fibrinolysis. In fact, hypo and basal fibrinolysis are always grouped together and for the same reasons we also kept together dogs with primary and secondary increased fibrinolysis and primary and secondary hyperfibrinolysis. These information are now better spelled/explained in the captions of tables 3, 4, 5, and 6.

Our cut-offs for the ROTEM and traditional parameters are not as “arbitrary” as you state. In fact, in part they are justify throughout the paper with biological explanations (i.e., fibrinogen) and in part they are in line with human and veterinary studies where fibrinolysis cut-offs are based on reference intervals. Nevertheless, we do agree with you that, as any new classification system, replication studies need to be performed to validate our proposed classification system. 

Finally, we cannot adhere to your suggestion of using a more simplified scheme including only the hyperfibrinolytic, the normal fibrinolytic and the hypofibrinolytic state for two reasons: 

a) It would change partially the target of our study. In fact, we also aim to try to differentiate patient with increased fibrinolysis that are at risk of bleeding (hyperfibrinolysis) from patient with increased fibrinolysis that are probably not at risk of bleeding (increased fibrinolysis). In addition, we also try to differentiate primary from secondary fibrinolysis. Unfortunately, we would not be able to do these types of differentiations with the scheme from you proposed. 

b) As a LI60 of 100% essentially encompasses the normal range, some authors consider the ROTEM in essence unsuitable to define hypofibrinolysis (although this is done extensively in human’s trauma research). For this reason, we decided in our inferential statistics to combine the hypo and the basal fibrinolysis in a single group, leaving this distinction only for the descriptive statistics. 

REFERENCE: Lisman T. Decreased Fibrinolytic Capacity in Cirrhosis and Liver Transplantation Outcomes. Liver Transpl. 2019 Mar;25(3):359-361. 

Reviewer #2: 

1) In this veterinary clinical study, the authors select two groups of animals from a large cohort considering the presence of cavitary effusion. Their aim is to study the fibrinolytic status of these animals in relation to the presence of pleural, peritoneal, or pericardial effusions by ROTEM technology. The study has several limitations that should be considered by the authors. They do not have a healthy control study group to compare their results on sick animals. The authors label their control group as "Sick dogs without intracavitary effusion". However, in both groups the animals are sick. It would be important to have an adequate healthy group for comparison.

Authors’ reply: To give an idea for comparison to a healthy population, reference intervals (obtained from healthy animals) for all the parameters studied were included (and still are) in the result section of our paper. On the other hand, a true healthy control group cannot be added, as requested from You, because it would heavily damage the validity of our study (please see below). 

Case–control is a type of epidemiological observational study where subjects are observed in order to determine both their exposure and outcome status. A common misconception among investigators is that case-control design is perceived as a comparison between a group of diseased individuals versus a group of “normal” (i.e., disease-free) individuals. This conceptualization of the case-control design is inconsistent with the actual goal of control group selection, which is to provide a representative sample of the risk factors distribution among the population at risk. Therefore, following the current epidemiology textbooks indications and international guidelines on observational studies, we have included as a control population a group of sick dogs as much as possible similar to the specific diseased group (cases), but without the specific outcome of interest that we wanted to study (i.e., the presence of intracavitary effusion). On the other hand, a control group of healthy animals would have weakened the study, because different for too many aspects from a diseased group and not even representative of the general population. Selection of healthy subjects has been heavily criticized in the modern literature and the so-called “healthy patient bias” has been coined (REFERENCES follow). 

1) Bloom MS, Schisterman EF, Hediger ML. Selecting controls is not selecting “normals”: design and analysis issues for studying the etiology of polycystic ovary syndrome. Fertil Steril. 2006;86(1):2-12

2) Lewallen S, Courtright P. Epidemiology in practice: case-control studies. Comm Eye Health 1998;11:57-58

3) Wacholder JK, Silverman JS, McLaughlin JK, Mandel JS. Selection of controls in case-control studies. II. Types of Controls. Am J Epidemiol; 1992, 135:1029-1041

4) Shrank WH, Patrick AR, Brookhart A. Healthy user and related biases in observational studies of preventive interventions: a primer for physicians. J Gen Intern Med, 2011;26(5):546-550

2) The presence of bleeding in a portion of the individuals in both groups complicates the interpretation. The authors should include an analysis of their results if these animals are excluded.

Authors’ reply: We thank the Reviewer for this comment that allow us to improve our article. To address your concern additional statistical analysis excluding bleedings dogs from both groups were performed as requested. The additional analyses showed that bleeding is not the solely driving of our finding. In fact, excluding these dogs none of the results changed. Please see 'Revised Manuscript with Track Changes’ at: 

• Materials and methods lines: 214-220

• Results: lines 269-271; 279-283; 357-360; 364-367; 372-374; Table 1, 3, 4, and 5.

• Discussion: 427–430; 462–462; 484–485; 520–521. 

6. PLOS authors have the option to publish the peer review history of their article (what does this mean?). If published, this will include your full peer review and any attached files.

Do you want your identity to be public for this peer review? For information about this choice, including consent withdrawal, please see our Privacy Policy.

Reviewer #1: No

Authors’ reply: none requested

Reviewer #2: No

Authors’ reply: none requested

---

## [Decision Letter · Decision Letter 1]

30 Sep 2019

PONE-D-19-15389R1

Enhanced fibrinolysis detection in a natural occurring canine model with intracavitary effusions: comparison and degree of agreement between thromboelastometry and FDPs, D-dimer and fibrinogen concentrations

PLOS ONE

Dear Mr Zoia,

Thank you for submitting your manuscript to PLOS ONE. After careful consideration, we feel that it has merit but does not fully meet PLOS ONE’s publication criteria as it currently stands. Therefore, we invite you to submit a revised version of the manuscript that addresses the points raised during the review process.

The authors have addressed the concerns adequatedly. For clarity, it would be interesting that some changes in the organization of the section are considered, as suggested by the reviewers.

We would appreciate receiving your revised manuscript by Nov 14 2019 11:59PM. To enhance the reproducibility of your results, we recommend that if applicable you deposit your laboratory protocols in protocols.io, where a protocol can be assigned its own identifier (DOI) such that it can be cited independently in the future. For instructions see: http://journals.plos.org/plosone/s/submission-guidelines#loc-laboratory-protocols

We look forward to receiving your revised manuscript.

Kind regards,

Pablo Garcia de Frutos

Academic Editor

PLOS ONE

Reviewers' comments:

Reviewer's Responses to Questions

**Comments to the Author**

1. If the authors have adequately addressed your comments raised in a previous round of review and you feel that this manuscript is now acceptable for publication, you may indicate that here to bypass the “Comments to the Author” section, enter your conflict of interest statement in the “Confidential to Editor” section, and submit your "Accept" recommendation.

Reviewer #1: (No Response)

Reviewer #2: All comments have been addressed

2. Is the manuscript technically sound, and do the data support the conclusions?

Reviewer #1: Yes

Reviewer #2: Yes

3. Has the statistical analysis been performed appropriately and rigorously? 

Reviewer #1: Yes

Reviewer #2: Yes

4. Have the authors made all data underlying the findings in their manuscript fully available?

Reviewer #1: Yes

Reviewer #2: Yes

5. Is the manuscript presented in an intelligible fashion and written in standard English?

Reviewer #1: Yes

Reviewer #2: Yes

6. Review Comments to the Author

Reviewer #1: Thank you for mostly addressing the major concerns raised in my previous review. I have a few additional minor comments that I would recommend the authors address before publication.

(1) Thank you for better describing the rationale behind the study regarding the potential contribution of effusions to systemic hyperfibrinolysis. Given that this is the main point of the study, I would recommend that you move that description from the Discussion to the Introduction.

(2) Your manuscript as it stands does not include any hypotheses. In the interest of informing the reader about the rationale for your statistical analyses, it would be appropriate to rewrite the final paragraph of your introduction to reframe the objectives of your study as specific, testable hypotheses that are linked back to your statistical analyses.

Best of luck with your manuscript.

Reviewer #2: The authors have answered the criticisms on the initial version of the manuscript. The laboratory reference intervals could be included in the relevant tables and figure.

7. PLOS authors have the option to publish the peer review history of their article (what does this mean?). If published, this will include your full peer review and any attached files.

Reviewer #1: No

Reviewer #2: No

---

## [Author Response · Author response to Decision Letter 1]

23 Oct 2019

RE: Manuscript PONE-D-19-15389R1

Enhanced fibrinolysis detection in a natural occurring canine model with intracavitary effusions: comparison and degree of agreement between thromboelastometry and FDPs, D-dimer and fibrinogen concentrations

Dear Dr. Pablo Garcia de Frutos, 

The authors would like to thank You and the Reviewers for the time spent and for their constructive comments in attempt to improve our manuscript. 

Few changes (listed below) have been made to the current manuscript to incorporate the suggestions of the Reviewers in the organization of the section of the manuscript. We hope that these are satisfactory to allow for publication. 

Kind regards,

Andrea Zoia, DVM, Cert SAM, Dip ECVIM-CA

(Corresponding Author)

PS: If you have any further questions or comments, please do not hesitate to contact me.

Comments to the Author

1. If the authors have adequately addressed your comments raised in a previous round of review and you feel that this manuscript is now acceptable for publication, you may indicate that here to bypass the “Comments to the Author” section, enter your conflict of interest statement in the “Confidential to Editor” section, and submit your "Accept" recommendation.

Reviewer #1: (No Response)

Authors’ reply: none requested

Reviewer #2: All comments have been addressed

Authors’ reply: none requested

2. Is the manuscript technically sound, and do the data support the conclusions?

Reviewer #1: Yes

Authors’ reply: none requested

Reviewer #2: Yes

Authors’ reply: none requested

3. Has the statistical analysis been performed appropriately and rigorously?

Reviewer #1: Yes

Authors’ reply: none requested

Reviewer #2: Yes

Authors’ reply: none requested

4. Have the authors made all data underlying the findings in their manuscript fully available?

Reviewer #1: Yes

Authors’ reply: none requested

Reviewer #2: Yes

Authors’ reply: none requested

5. Is the manuscript presented in an intelligible fashion and written in standard English?

Reviewer #1: Yes

Authors’ reply: none requested

Reviewer #2: Yes

Authors’ reply: none requested

6. Review Comments to the Author

Reviewer #1: Thank you for mostly addressing the major concerns raised in my previous review. I have a few additional minor comments that I would recommend the authors address before publication.

(1) Thank you for better describing the rationale behind the study regarding the potential contribution of effusions to systemic hyperfibrinolysis. Given that this is the main point of the study, I would recommend that you move that description from the Discussion to the Introduction.

Authors’ reply: The relevant paragraphs has been moved from the discussion ('Revised Manuscript with Track Changes’ at lines: 442 - 446) to the introduction as requested ('Revised Manuscript with Track Changes’ at lines: 63 - 68) 

.

(2) Your manuscript as it stands does not include any hypotheses. In the interest of informing the reader about the rationale for your statistical analyses, it would be appropriate to rewrite the final paragraph of your introduction to reframe the objectives of your study as specific, testable hypotheses that are linked back to your statistical analyses.

Best of luck with your manuscript.

Authors’ reply: The final paragraph has been re-written stating our 3 hypotheses of the study. i.e.: 

1) ROTEM could detect an enhanced and more severe pattern of fibrinolysis in dogs with intracavitary effusions compared to dogs without effusion. 

2) There was an agreement in fibrinolysis severity detection between the combination of ROTEM assay results and fibrinogen concentrations, or alternatively by concentrations of FDPs, D-dimers and fibrinogen. 

3) Dogs with intracavitary effusions had more primary fibrinolysis/hyperfibrinolysis compared to dogs without effusion when assessed by concentrations of FDPs, D-dimers and fibrinogen.

Please see the 'Revised Manuscript with Track Changes’ at lines: 82 – 95.

Reviewer #2: The authors have answered the criticisms on the initial version of the manuscript. The laboratory reference intervals could be included in the relevant tables and figure.

Authors’ reply: The laboratory reference intervals have been included in the relevant tables and figure as requested. Please see 'Revised Manuscript with Track Changes’ at: table 1 and 2 + Fig 1. 

7. PLOS authors have the option to publish the peer review history of their article (what does this mean?). If published, this will include your full peer review and any attached files.

Do you want your identity to be public for this peer review? For information about this choice, including consent withdrawal, please see our Privacy Policy.

Reviewer #1: No

Authors’ reply: none requested

Reviewer #2: No

Authors’ reply: none requested

---

## [Editor Report · Decision Letter 2]

30 Oct 2019

Enhanced fibrinolysis detection in a natural occurring canine model with intracavitary effusions: comparison and degree of agreement between thromboelastometry and FDPs, D-dimer and fibrinogen concentrations

PONE-D-19-15389R2

Dear Dr. Zoia,

We are pleased to inform you that your manuscript has been judged scientifically suitable for publication and will be formally accepted for publication once it complies with all outstanding technical requirements.

With kind regards,

Pablo Garcia de Frutos

Academic Editor

PLOS ONE
---

## [Editor Report · Acceptance letter]

8 Nov 2019

PONE-D-19-15389R2 

Enhanced fibrinolysis detection in a natural occurring canine model with intracavitary effusions: comparison and degree of agreement between thromboelastometry and FDPs, D-dimer and fibrinogen concentrations 

Dear Dr. Zoia:

I am pleased to inform you that your manuscript has been deemed suitable for publication in PLOS ONE. Congratulations! Your manuscript is now with our production department. 

With kind regards,

on behalf of

Dr. Pablo Garcia de Frutos 

Academic Editor

PLOS ONE